# Efficient stabilization of cyanonaphthalene by fast radiative cooling and implications for the resilience of small PAHs in interstellar clouds

**Mark H. Stockett** [ORCID][1] ✉, **James N. Bull** [ORCID][2], **Henrik Cederquist** [ORCID][1], **Suvasthika Indrajith**[1], **MingChao Ji** [ORCID][1], **José E. Navarro Navarrete**[1], **Henning T. Schmidt** [ORCID][1], **Henning Zettergren** [ORCID][1] & **Boxing Zhu**[1]

After decades of searching, astronomers have recently identified specific Polycyclic Aromatic Hydrocarbons (PAHs) in space. Remarkably, the observed abundance of cyanonaphthalene (CNN, $C_{10}H_7CN$) in the Taurus Molecular Cloud (TMC-1) is six orders of magnitude higher than expected from astrophysical modeling. Here, we report unimolecular dissociation and radiative cooling rate coefficients of the 1-CNN isomer in its cationic form. These results are based on measurements of the time-dependent neutral product emission rate and kinetic energy release distributions produced from an ensemble of internally excited 1-CNN$^+$ studied in an environment similar to that in interstellar clouds. We find that Recurrent Fluorescence – radiative relaxation via thermally populated electronic excited states – efficiently stabilizes 1-CNN$^+$, owing to a large enhancement of the electronic transition probability by vibronic coupling. Our results help explain the anomalous abundance of CNN in TMC-1 and challenge the widely accepted picture of rapid destruction of small PAHs in space.

Polycyclic Aromatic Hydrocarbons (PAHs) have long been thought to be ubiquitous in the Interstellar Medium (ISM), as evidenced by the infrared (IR) emission bands observed by astronomers at wavelengths coincident with their vibrational transition energies[1]. These partially resolved bands, however, are common to PAHs as a class of molecules and cannot be used to identify specific species. To date, it has been generally held that PAHs in the interstellar medium must be fairly large, containing more than 50 carbon atoms[1], to be resilient against fragmentation after collisions or photon absorption. Here, we present experimental results demonstrating that a small, vibrationally hot PAH cation is stabilized much more rapidly than previously assumed, and that this occurs efficiently through Recurrent Fluorescence (RF).

Recently, McGuire et al.[2] analyzed radio telescope observations of the dark molecular cloud TMC-1 to identify two isomers of cyanonaphthalene ($C_{10}H_7CN$, Fig. 1) – in which a nitrile/cyano (-CN) group replaces one of the hydrogen atoms of naphthalene. This is the first definitive assignment of a specific PAH molecule in space. As important as this assignment is for finally confirming the presence of PAH molecules in space, it is equally remarkable that the observed CNN abundances are six orders of magnitude higher than predicted by astrochemical modeling[2]. Notably, the same model generally reproduces the observed abundances of linear and mono-cyclic nitriles[3], but underpredicts the abundance of bicyclic indene by four orders of magnitude[4]. Laboratory studies have shown that nitrogen bearing

---

[1]Department of Physics, Stockholm University, Stockholm, Sweden. [2]School of Chemistry, University of East Anglia, Norwich, United Kingdom.
✉e-mail: Mark.Stockett@fysik.su.se

**Fig. 1 | Found in space.** Molecular structures of 1-cyanonaphthalene (1-CNN, left) and 2-CNN (right), $C_{10}H_7CN$.

aromatic molecules[5], as well as pure hydrocarbons like naphthalene[6] and indene[4,7], may be readily formed in barrierless, gas-phase ion-molecule and radical-molecule reactions under conditions similar to those in interstellar clouds. On the other hand, the model of McGuire et al. indicates rapid depletion of CNN from TMC-1 through interactions with ions,[3] and it predicts a limited abundance of its main precursor, naphthalene, which has also been shown to be a keystone in the formation of larger PAHs[8]. McGuire et al. argue that radiative cooling of small PAH cations ( < 20 atoms) is too slow to stabilize the molecules following ionizing interactions.[3] However, this reasoning is based on the assumption that cooling only occurs via emission of IR photons from transitions between vibrational levels, which is indeed slow[9,10]. However, faster cooling may be possible due to emission of optical photons from thermally populated electronically excited states, i.e. Recurrent Fluorescence[11,12], also called Poincaré fluorescence.

Investigations of Recurrent Fluorescence are an emerging theme of research within laboratory astrophysics. In a typical experiment, internally hot ions are produced in a plasma ion source or through laser excitation followed by internal conversion to give the vibrationally-excited electronic ground state[13,14]. These hot ions may, by inverse internal conversion, spontaneously[15] populate electronically excited states which may, in turn, relax by emitting optical photons. While a few outstanding experiments have succeeded in direct detection of RF photons[14,16], most reports of RF have been inferred indirectly through the quenching of dissociation or other destruction channels on timescales that are too rapid to be explained by sequential emission of IR photons[13,17,18]. Indirect determinations of RF rates depend sensitively on the absolute rate coefficient of the monitored destruction channel, which often have been calculated using simple statistical models[18,19] or determined through separate mass spectrometry experiments with much shorter sampling times[12–14,20]. Additionally, it is not uncommon for published rate coefficients for the same molecule and destruction channel to differ by orders of magnitude[12,21].

Here, we present a method to determine unimolecular dissociation and RF quenching rate coefficients within the same experiment. Using a cryogenic ion beam storage ring providing 'molecular cloud in a box' conditions, we show that cooling is much faster than assumed by McGuire et al. Thus, some of the destruction channels included in the modelling by McGuire et al.[2] are avoided, leading to a significantly higher survival probability of 1-CNN⁺ in interstellar clouds. The dominant dissociation channel of 1-CNN⁺ observed under our experimental conditions is[22]:

$$C_{10}H_7CN^+ \rightarrow C_{10}H_6{}^+ + HCN + \epsilon, \qquad (1)$$

where $\epsilon$ is the kinetic energy released in the reaction. Time-dependent dissociation rates $\Gamma(t)$ and Kinetic Energy Release (KER) distributions $P(\epsilon, t)$ were measured for ensembles of 1-CNN cations produced with an initially broad internal energy distribution, $g(E, t = 0)$, leaving the ion source at time $t = 0$. The ensemble was stored in the cryogenic electrostatic ion-beam storage ring DESIREE[23,24] (see Sec. 4.1). From analysis of the KER distributions, we determine the absolute unimolecular dissociation rate coefficient, $k_{diss}(E)$, as a function of the internal excitation energy, $E$. We reproduce the measured absolute

**Fig. 2 | Time-dependent dissociation rate of ensembles of internally hot 1-CNN⁺.** The storage time, $t$, is relative to formation of the ions in the ion source. $R(t)$ is the rate of neutral product detection recorded continuously during storage of the ion beam. The dotted curve 'Fit' is a fit of Equation (2) to $R(t)$ with a constant background added. $\Gamma(t)$ is the absolute, ensemble-averaged dissociation rate per molecule. '$\Gamma(t)$ Img.' is the rate extracted from KER distributions measured at specific times. The error bands (shaded areas) and error bars are the standard deviations. The solid curve 'Sim.' is the result of our master equation simulation of $\Gamma(t)$ including Herzberg-Teller coupling, which gives $f = 0.011$, and the best-fitting initial temperature of 1970 K. The dashed curve 'Sim. (No HT)' is a simulation without Herzberg-Teller coupling ($f = 10^{-4}$) with a best-fitting initial temperature of 1360 K.

dissociation rate, $\Gamma(t)$, with master equation simulations (Sec. 4.2) of the time evolution of the energy distribution, $g(E, t)$, and calculated vibrational and electronic (RF) cooling rate coefficients. These simulations only reproduce $\Gamma(t)$ when we include Herzberg-Teller vibronic coupling in the RF rate coefficient. The implications of these results for the observed high abundance of 1-CNN in TMC-1 are discussed in Sec. 3.

## Results

### Dissociation rates

The measured dissociation rate for ensembles of internally hot 1-CNN cations stored in DESIREE is shown in Fig. 2. During the measurement, the yield of neutral fragments leaving the storage ring is recorded continuously as a function of time $t$ after the ions left the source at $t = 0$. The measured count rate, $R(t)$, is averaged over a large number of injection and storage cycles. We assume that $R(t)$ is dominated by dissociation of 1-CNN⁺ yielding HCN molecules according to Equation (1). The products of competing dissociation channels are not expected to contribute significantly to $R(t)$, and are discussed in Supplementary Note 1. During the first $10^{-3}$ seconds after formation, the count rate follows a power law $R(t) \propto t^{-1}$, as a consequence of the broad distribution $g(E, t)$ of internal energies, $E$, and rapid variation with $E$ of the dissociation rate coefficient, $k_{diss}(E)$[19]. After a critical time $k_c^{-1}$, $R(t)$ is quenched by competition with radiative cooling[25], giving an approximate time dependence:

$$R(t) = r_0 t^{-1} e^{-k_c t}. \qquad (2)$$

The dotted curve labeled 'Fit' in Fig. 2 is a fit of this equation, with a constant background term added, to the measured data $R(t)$. The critical rate coefficient, where the dissociation and radiative rate coefficient are roughly equal, is determined to be $k_c = 300(20)$ s⁻¹, and $r_0 = 106.55(13)$. The value of $k_c$ alone does not provide any information on the cooling mechanism or the internal energy $E_c$ at which $k_{diss}(E_c) \approx k_c$, which is the relevant quantity for predicting the survival

probability of an excited 1-CNN$^+$ molecule in space. For this, the absolute dissociation rate coefficient needs to be determined as a function of $E$.

The absolute per-particle dissociation rate is the ensemble average of the dissociation rate coefficient:

$$\Gamma(t) = \int g(E,t)k_{\text{diss}}(E)dE \Big/ \int g(E,t)dE. \qquad (3)$$

and is related to the measured count rate, $R(t)$, by[20,24]

$$\Gamma(t) = \frac{C}{\eta_{\text{det}}L_{\text{SS}}N(t)}R(t), \qquad (4)$$

where $\eta_{\text{det}} = 0.34(3)$ (see Supplementary Methods) is the efficiency for detection of HCN, $C = 8.7$ m is the circumference of the storage ring, $L_{\text{SS}} = 0.95$ m is the length of the stored beam viewed by the detector, and $N(t)$ is the average number of stored ions remaining in the ring at time $t$. The latter is determined from the count rate $R(t)$, measured during ion storage, and the terminal ion beam current, measured at the end of each injection-storage cycle. The absolute dissociation rate $\Gamma(t)$ for 1-CNN$^+$ is given in Fig. 2, with the background count rate due to collisions with residual gas ($0.23(1)$ s$^{-1}$) and detector dark noise ($6.78(2)$ s$^{-1}$) subtracted from the experimental data.

### Kinetic energy release distributions

In order to connect the dissociation rate $\Gamma(t)$ to the vibrational energy $E$, we analyze the KER distributions to determine the unimolecular dissociation rate coefficient in Arrhenius form:

$$k_{\text{diss}}(E(T^{\ddagger})) = A^{\text{diss}}e^{-E_a/k_B T^{\ddagger}}, \qquad (5)$$

where $E_a$ and $A^{\text{diss}}$ are the activation energy and pre-exponential factor of the decay channel, respectively, and $T^{\ddagger}$ is the temperature of the transitory ion-molecule complex. The latter is related to the vibrational energy $E$ through the caloric curve (see Supplementary Fig. 1). In transition state theory, the pre-exponential factor is given by

$$A^{\text{diss}} = \frac{k_B T^{\ddagger}}{h}e^{1+\frac{\Delta S^{\ddagger}}{N_A k_B}}, \qquad (6)$$

where $\Delta S^{\ddagger}$ is the activation entropy and $N_A$ is Avogadro's number[26].

The KER distribution measured for the ensemble of ions at $t = 120$ $\mu$s is shown in Fig. 3A. This KER distribution is well-reproduced by the model elaborated by Hansen[27], which considers a transition state with activation energy $E_a$, reverse barrier height $\Delta E$, and a potential near the saddle point $V(z) \approx \Delta E - \frac{1}{2}\mu\omega^2 z^2$, where $z$ is the reaction coordinate measured from the top of the barrier, $\mu$ is the reduced mass of the dissociation products (Equation (1)), and $\mu\omega^2$ is the radius of curvature of the potential near the saddle point. The model parameters are illustrated schematically in the inset to Fig. 3A. The KER distribution takes the form[27]:

$$P(\epsilon) \propto \frac{e^{\beta'}}{e^{\beta'}+1}e^{-(\epsilon-\Delta E)/k_B T^{\ddagger}},$$
$$\text{where } \beta' = 4\pi\frac{\Delta E}{h\omega}\left(\sqrt{\frac{\epsilon}{\Delta E}}-1\right). \qquad (7)$$

The solid line in Fig. 3A is a fit of Equation (7), giving a temperature $T^{\ddagger} = 1610(20)$ K. The reverse barrier $\Delta E = 6.2(5)$ meV, also obtained from the fit, is comparable to the centrifugal barrier for 1-CNN assuming the rotational temperature is of the same order as $T^{\ddagger}$[27]. Given such a 'flat' transition state[26], we invoke the simplifying assumption $1 + \frac{\Delta S^{\ddagger}}{N_A k_B} \approx 0$ (see Supplementary Note 2),

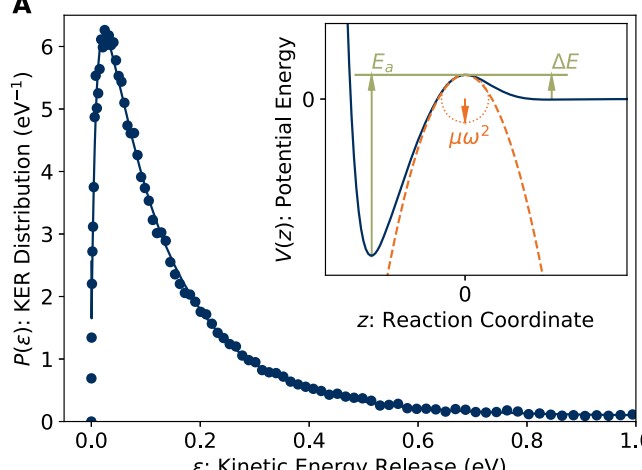

**A**

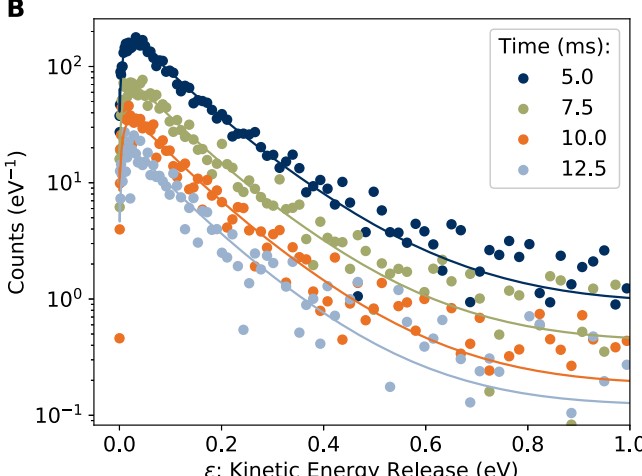

**B**

**Fig. 3 | Kinetic Energy Release (KER) distributions for HCN-loss from 1-CNN$^+$.** **A** Normalized KER distribution recorded 120 $\mu$s after ion formation. The solid line is a fit of Equation (7). The inset illustrates the model parameters. **B** KER distributions recorded 5.0–12.5 ms after ion formation. Solid lines are the results of a simultaneous fit to the KER distributions from 120 $\mu$s to 27.5 ms. Time refers to the beginning of the $\Delta t = 1.1$ ms camera exposure. Note the logarithmic vertical scale.

giving:

$$k_{\text{diss}}(E(T^{\ddagger})) \approx \frac{k_B T^{\ddagger}}{h}e^{-E_a/k_B T^{\ddagger}}, \qquad (8)$$

where the error introduced in $k_{\text{diss}}$ due to this approximation is much smaller than the error due to the uncertainty in $E_a$. We next simultaneously fit the KER distributions $P(\epsilon,t)$ recorded up to $t = 27.5$ ms, subject to the constraint:

$$\int P(\epsilon,t)d\epsilon = R(t)\Delta t = r_0 k_{\text{diss}}(E(T^{\ddagger}))\Delta t \qquad (9)$$

where $\Delta t$ is the camera exposure time, and $k_{\text{diss}}(E(T^{\ddagger}))$ is given by Equation (8). The value of $r_0$ is taken from the fit to $R(t)$, which is to assume that $R(t)$ is dominated by the modeled HCN-loss channel (see Supplementary Note 1). The values of $\Gamma(t)$ extracted from the imaging data (Equation (9)) are shown in Fig. 2, labeled '$\Gamma(t)$ Img.', and agree well with those from the decay rate measurement (Equation (4)), supporting this assumption. The fit of $P(\epsilon,t)$ includes contributions from ions dissociating at different points along the straight section of

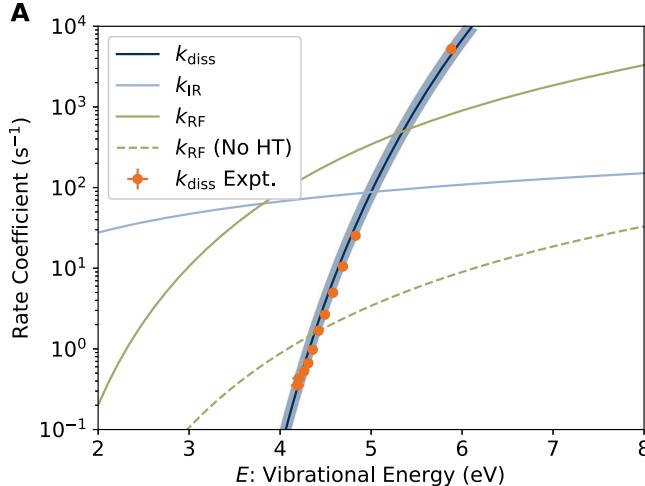

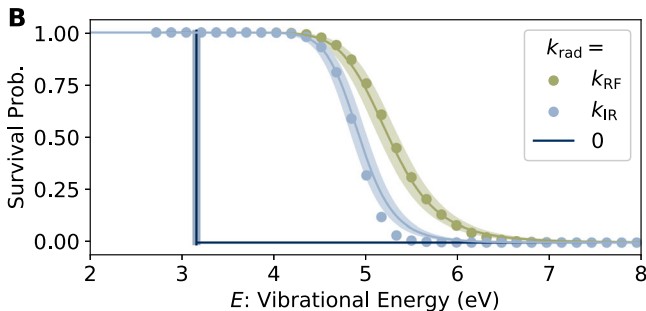

**Fig. 4 | Rate coefficients and survival probabilities. A** Dissociation rate coefficient, $k_{diss}(E)$, fit to the experimental values derived from KER distributions. The error bars are standard deviations of the fit parameter estimates. **B** Survival probability for 1-CNN$^+$ from master equation simulations (symbols) and from the approximate expression $k_{rad}/(k_{diss} + k_{rad})$ (solid lines). Error bands (shaded areas) in both panels reflect the uncertainty due to the standard deviation in the estimate of $E_a$.

the storage ring seen by the detector. Results for $t = 5.0$, 7.5, 10.0, and 12.5 ms are shown in Fig. 3B. The fitted temperatures for these $P(\epsilon, t)$ distributions range from 1320(10) to 1220(10) K. From the full data set, including $t = 120$ μs, we obtain $E_a = 3.16(4)$ eV. Our results are close to those of So & Dunbar[28] for benzonitrile (which is structurally similar to CNN), who determined $E_a = 3.015$ eV, and of West et al.[22] who estimated an activation energy "from 2.5 to 3 eV" for 1- and 2-CNN.

Our measured dissociation rate coefficients from the imaging experiments are plotted in Fig. 4, along with the dissociation rate coefficient in microcanonical form according to the inverse Laplace transform formula[12]:

$$k_{diss}(E) = A_{1000K}^{diss} \frac{\rho(E - E_a)}{\rho(E)}, \qquad (10)$$

where $\rho(E)$ is the vibrational level density of 1-CNN$^+$, and we adopt the nominal value of the pre-exponential factor $A_{1000K}^{diss} = k_B(1000 \text{ K})/h = 2 \times 10^{13}$ s$^{-1}$. The transition state temperatures $T^{\ddagger}$ have been converted to vibrational energies $E$ according to the caloric curve computed directly from the vibrational frequencies[29] and including the finite heat bath correction[25] (see Supplementary Fig. 1).

### Radiative stabilization

We model the unimolecular dissociation rate $\Gamma(t)$ in Fig. 2 with the master equation approach described in Section 4.2. The calculated RF and IR cooling rate coefficients are plotted in Fig. 4. Assuming the initial vibrational energy distribution is approximated by Boltzmann

statistics, we find that an initial temperature of 1970(60) K most closely reproduces the experimental dissociation rate. The simulated dissociation rate is given by the solid line labeled 'Master' in Fig. 2. The simulated curve deviates from the experimental data for $t > 20$ ms, which may be attributed to sequential fragmentation processes that have been observed in other ion-beam storage experiments with PAH ions[20].

The quenching of the dissociation rate is consistent with Recurrent Fluorescence from the lowest electronic excited state, $L_\alpha$. To reproduce the measured dissociation rate, consideration of Herzberg-Teller (H-T) vibronic coupling proved essential. This transition, unlike the first electronic transitions of highly symmetric PAHs[30], is not strictly forbidden, but is still quite weak in the Franck-Condon limit. Including H-T coupling, the calculated transition oscillator strength, $f = 0.011$, and hence RF rate coefficient (Equation (14)) is two orders of magnitude higher than if vibronic coupling is neglected (see Sec. 4.2). A simulation with the lower oscillator strength ($f = 10^{-4}$) is shown by the dashed line labeled 'Master (No HT)' in Fig. 2 for the best-fitting initial temperature of 1360(20) K. These findings are similar to those of a previous study of perylene cations[20], which reported an RF rate consistent with an emission oscillator strength of 0.05(1), whereas most calculated values neglecting vibronic coupling are on the order of $10^{-4}$. The vibronic enhancement of $f$ in 1-CNN$^+$ is likely commonplace for a broad range of PAH molecules and may be decisive for the stabilization of such molecules in cold interstellar environments – in stark contrast to what has been assumed previously.

The modeled survival probability of vibrationally hot 1-CNN$^+$ is plotted as a function of vibrational energy $E$ in Fig. 4B. The data points are determined from master equation simulations initialized to $\delta$-function vibrational energy distributions. Also included are simulations in which the RF rate coefficient is artificially set to zero, leaving only IR cooling. The solid line is the branching ratio $k_{rad}/(k_{diss} + k_{rad})$, where $k_{rad} = k_{RF} + k_{IR}$ or $k_{IR}$. The vertical step indicates the dissociation threshold ($k_{rad} = 0$). The branching ratio when $k_{rad} = k_{RF} + k_{IR}$ agrees well with the full simulation, implying that a single RF photon stabilizes 1-CNN$^+$[31]. In contrast, when $k_{rad} = k_{IR}$, the branching ratio overestimates the survival probability from the simulations, as multiple IR photons are required to quench the decay. We conclude that while both cooling modes stabilize 1-CNN$^+$ ions with internal energies well above the dissociation threshold, RF does so much more efficiently for a critical range of internal excitation energies, as will be discussed below.

## Discussion

While it has long been known that Herzberg-Teller vibronic coupling influences the optical spectra of PAHs[32], the implications are not widely incorporated in astronomical contexts. For example, comparisons of measured and simulated optical spectra of PAHs to astronomical observations such as the Diffuse Interstellar Bands often discount the importance of the lowest energy transitions, on the basis of their low oscillator strengths in the Franck-Condon limit. In the present case, the large enhancement of the oscillator strength of the weak $L_\alpha \leftarrow D_0$ transition leads to efficient radiative stabilization of 1-CNN$^+$ by RF. Generally, H-T coupling and RF are both important for PAHs[13,14,33,34], and the possibility of fast radiative cooling challenges the long-held assumption that small PAHs are rapidly depleted from the ISM by UV photodestruction[9,10]. The importance of RF in stabilizing a particular PAH depends sensitively on the absolute dissociation rate coefficients and optical transition probabilities. Calculations of these properties, including H-T coupling, benchmarked to quantitative laboratory data such as those presented here, will be crucial to predicting the stability of PAHs and other complex molecules in space.

In the specific case of CNN in TMC-1, the present results suggest that the abundance predicted by astrochemical models may be too low due to both unrealistically low formation rates and unrealistically high destruction rates in the model[2]. McGuire et al. acknowledge that small

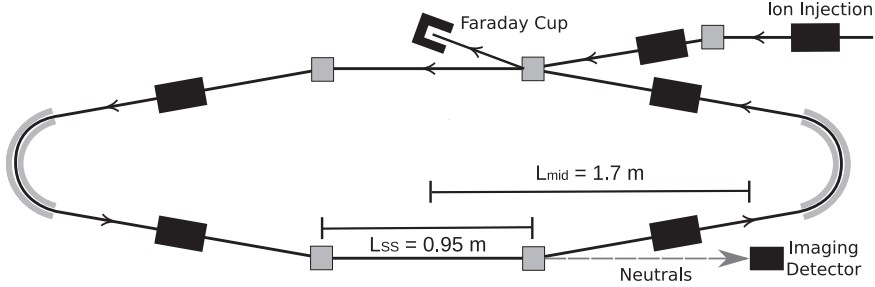

**Fig. 5 | Schematic of the DESIREE electrostatic ion storage ring.** Beams of 1-CNN$^+$ with kinetic energies of 34 keV circulate along the trajectory indicated by the solid lines with arrows. Neutral products formed in the lower straight section follow the dashed line and are counted with the Imaging Detector.

PAHs, such as naphthalene, may be inherited from the diffuse ISM at early stages of the evolution of TMC-1, but that formation of 1-CNN from these inherited PAHs is disfavored due to the inefficiency of radiative stabilization for small PAHs through IR-emission. However, RF is known to be an important mode of radiative cooling for the small unsubstituted PAH cations naphthalene[14], anthracene[13], and perylene[20], and is almost certainly more important for PAHs in general than predicted by calculations neglecting Herzberg-Teller coupling. The inherited abundances of small PAHs from the diffuse ISM required to explain the observed amounts of CNNs in TMC-1 may thus not be as "unrealistically large" as previously thought[2].

The astrochemical model[2] employed by McGuire et al. likely overestimates the rate of destruction of 1-CNN in TMC-1. In the model[2], naphthalene and CNNs are primarily destroyed in reactions with C$^+$, H$^+$, He$^+$, H$_3^+$, and H$_3$O$^+$ leading to small linear fragments such as C$_2$H$_4$. However, some of the reactions included can be expected to lead predominantly to charge transfer rather than fragmentation[35,36], e.g.

$$C^+ + C_{10}H_7CN \rightarrow C + C_{10}H_7CN^+ \tag{11}$$

For this example, the excess energy of the reaction, i.e. the 3.2 eV difference in ionization potentials[37], is insufficient to dissociate 1-CNN$^+$ at interstellar temperatures. The analogous reaction with H$^+$ (5.5 eV excess energy) should lead primarily, according to the present study, to HCN-loss (Equation (1)) and RF-stabilized 1-CNN$^+$, rather than disintegration into linear fragments. While 1-CNN$^+$ might be destroyed in dissociative recombination with electrons, radiative stabilization of the neutralized 1-CNN is possible[38]. Another process could involve 1-CNN reformation by mutual neutralization (MN) of 1-CNN$^+$ with PAH anions, which have been suggested to be the dominant negative charge carriers in dark clouds[39]. MN is also possible with linear hydrocarbon anions like C$_6$H$^-$, which have been identified in TMC-1[40] and have high electron binding energies[40], reducing the excess energy of the reaction. Finally, the proton-transfer reactions included in the model[2] of McGuire et al., e.g. with H$_3^+$, might lead to protonated PAHs, which are known to be stable and non-reactive[41], or to dehydrogenation, rather than carbon backbone fragmentation.

While photodestruction of 1-CNN is not part of the model of McGuire et al., we note that given the 8.6 eV ionization energy of 1-CNN[37] and efficient radiative stabilization of 1-CNN$^+$ by RF up to a critical vibrational energy $E_c \approx 5$ eV, neutral 1-CNN should be resilient against dissociative ionization by 13.6 eV recombination radiation, which dominates the radiation field of molecular clouds[42].

In summary, laboratory studies, including the present one, suggest that the underprediction of the CNN abundance in TMC-1 may be partially rationalized by a combination of both of the alternative scenarios considered by McGuire et al.: high inherited naphthalene abundance and no destruction by ions[2]. In their model, each of these assumptions individually reduced the discrepancy by about two orders of magnitude[2], although it should be noted that the assumed inherited naphthalene abundance was on the order of 1% of the available carbon.

Additional laboratory studies of ion-neutral, electron-ion, and ion-ion reactions involving CNN, as well as its precursors and fragmentation products[43], under astrophysically relevant conditions should be undertaken to further constrain astrochemical models.

From the groundbreaking study of McGuire et al. we now know from direct spectroscopic observations that a specific small PAH, 1-CNN, is present in the dark interstellar cloud TMC-1. In the present study we have shown that Recurrent Flourescence is sufficiently fast (due to Herzberg-Teller vibronic coupling) to stabilize ionized 1-CNN molecules and that they may survive in harsh astrophysical environments, making their large presence in TMC-1 easier to understand. As similarly fast radiative cooling processes occur in a range of small PAHs[13,14,20], it appears that the long-held assumption that small PAHs cannot survive in space has to be reconsidered.

## Methods

### Experiments

Experiments were conducted at the DESIREE (Double ElectroStatic Ion Ring ExpEriment) infrastructure at Stockholm University[23]. Cryogenic cooling of the DESIREE storage ring, which is shown schematically in Fig. 5, to ≈ 13 K results in a residual gas density on the order of ~10$^4$ cm$^{-3}$, consisting mostly of H$_2$[24].

1-CNN (Sigma-Aldrich, > 96%) was sublimed from powder in a resistively heated oven coupled to an electron cyclotron resonance (ECR) ion source (Pantechnik Monogan) using helium as a support gas. Cations extracted from the source were accelerated to 34 keV kinetic energy. Mass-selected beams of cationic 1-CNN$^+$ ($m/z = 153$) were stored in the DESIREE ion storage ring illustrated in Fig. 5.

After ion injection into DESIREE, neutral fragments are emitted from ions that retain sufficient internal energy from their formation process in the ion source. Neutrals formed in the observation arm (lower straight section in Fig. 5) of the storage ring continue with high velocity towards the position-sensitive Imaging Detector[44], which utilizes custom ultra-high dynamic range micro-channel plates (MCPs, Photonis) that are suitable for high count rates at cryogenic temperatures. Electrons produced when neutral fragments strike the MCP are converted to optical photons with a phosphor screen. The resulting images are recorded through a vacuum window using a CMOS camera (Photon Focus MV1-D2048).

Two types of neutral particle imaging experiment were performed. Firstly, a measurement was conducted with a continuous beam of 1-CNN$^+$ ions making a single pass around the ring. Including the transit time from the ion source to the storage ring, the ions in this experiment are decaying 120–124 $\mu$s after formation. For this single-pass measurement, a 0.5 mm aperture was inserted before the straight section of the storage ring to reduce the smearing of the neutral distributions due to the spatial extent of the beam. Secondly, measurements were performed using a stored beam circulating for 200 ms; no apertures were used. Exposures of $\Delta t = 1.1$ ms duration were triggered every 2.5 ms. The imaging data were time-stamped and synchronized with the storage cycle, allowing for analysis of the time-dependence of

the neutral particle distribution. Three-dimensional Newton spheres were reconstructed by applying an inverse Abel transform, using the 'three-point' algorithm implemented in the PyAbel package[45]. The density distribution is related to the KER distribution by:

$$\epsilon(r_{3D}) = \frac{m_{neut}}{m_{cat}} E_{acc} \left(\frac{r_{3D}}{L}\right)^2 \qquad (12)$$

where $\epsilon(r_{3D})$ is the KER associated with a radial slice of the Newton sphere of radius $r_{3D}$, $m_{neut}$ and $m_{cat}$ are the masses of the neutral and cationic reaction products, $E_{acc} = 34$ keV is the beam energy, and $L$ is the distance traveled by the products from the point of reaction to the detector. For clarity of presentation, the KER distributions in Fig. 3 are plotted against an $\epsilon$ scale calculated according to Equation (12) with $L = L_{mid}$, where $L_{mid} = 1.7$ m is the distance from the detector to the mid-point of the observation arm. Our analysis accounts for dissociation occurring along the full length of the observation arm by summing contributions to the Newton sphere density distribution in the detector plane from points at distances in the range $L_{mid} \pm L_{SS}/2$, where $L_{SS} = 0.95$ m is the length of the straight section seen by the detector (see Fig. 5). In the present case, the procedure gives a nearly insignificant correction relative to assuming all decays occur at $L_{mid}$.

The analysis of the dissociation count rate and KER distributions considered only HCN-loss (Equation (1)). At the energies relevant to this study, H-loss is the only other competitive dissociation pathway, with $C_2H_2$-loss being a minor channel[22]. Low-mass H atoms emitted with kinetic energies around 200 eV are detected very inefficiently by the MCP detectors[20] and are thus not expected to significantly contribute to the observed count rate, $R(t)$, or KER distributions. H-loss does deplete the stored ion beam and thus the per-particle dissociation rate $\Gamma(t)$.

## Kinetic Model

The following kinetic model for dissociation and radiative cooling of 1-CNN$^+$ is similar to those used in earlier studies of unsubstituted PAHs[20,46,47]. The dissociation rate coefficient, $k_{diss}(E)$, is modeled using Equation (10). The vibrational level density, $\rho(E)$, is computed using the Beyer-Swinehart algorithm[48]. Vibrational frequencies are calculated at the B3LYP/6-31G(d,p) level of Density Functional Theory (DFT) as implemented in Gaussian 16[49].

The infrared radiative (vibrational) cooling rate coefficient $k_{IR}$ is calculated within the Simple Harmonic Cascade (SHC) approximation[18]:

$$k_{IR}(E) = \sum_s k_s = \sum_s A_s^{IR} \sum_{\nu=1}^{\nu \leq E/h\nu_s} \frac{\rho(E - \nu h\nu_s)}{\rho(E)}, \qquad (13)$$

where $\nu$ is the vibrational quantum number, and $h\nu_s$ and $A_s$ are the transition energy and Einstein coefficient of vibrational mode $s$, respectively. Previous studies have shown the infrared cooling rates predicted by this model reproduce experimental data to within a factor of two[20,29].

The rate coefficient for RF is calculated using the expression[12]:

$$k_{RF}(E) = A^{RF} \frac{\rho(E - h\nu_{el})}{\rho(E)}, \qquad (14)$$

where the electronic transition energies, $h\nu_{el}$, and Einstein coefficients given by

$$A^{RF} = \frac{2\pi\nu_{el}^2 e^2}{\epsilon_0 m_e c^3} f, \qquad (15)$$

where $f$ is the oscillator strength, are taken from density functional theory. Because PAHs are prototype examples of Herzberg-Teller

activity[33,50] – i.e. coupled electronic and nuclear motion – the oscillator strength for RF ($L_\alpha$ band fluorescence) was modeled to be $f = 0.011$ using a Franck-Condon-Herzberg-Teller simulation[51] at the $\omega$B97X-D/cc-pVDZ level of theory. Here, higher level EOM-CCSD/cc-pVDZ calculations of the excitation energy $h\nu_{el}$ were performed in CFOUR[52]. The transition energy for RF is calculated to be $h\nu_{el} = 1.10$ eV at the equilibrium geometry of the lowest-lying $L_\alpha$ excited state.

The vibrational energy distribution, initially normalized such that $\int g(E, t = 0)dE = 1$, was propagated according to the Master Equation:

$$\begin{aligned}
\frac{d}{dt}g(E,t) = &-k_{diss}(E)g(E,t)/Y_{HCN} \\
&+ \sum_s [k_s(E + h\nu_s)g(E + h\nu_s, t) - k_s(E)g(E,t)] \\
&+ k_{RF}(E + h\nu_{el})g(E + h\nu_{el}, t) - k_{RF}(E)g(E,t).
\end{aligned} \qquad (16)$$

The first term gives the depletion of the population by unimolecular dissociation. The factor $Y_{HCN} = 0.70$ is the branching fraction for HCN-loss estimated from published breakdown curves[22] and accounts for depletion of the ion beam by H-loss. The first term in square brackets represents $\nu + 1 \to \nu$ vibrational emission from levels above $E$, while the second is $\nu \to \nu - 1$ emission to levels below $E$. The final two terms account for RF. The time step $dt$ is chosen to match the experimental data, with 32 extra points prior to the first experimental time bin to allow for the ion transit time from the ion source to the storage ring. The simulated dissociation rate is given by $\Gamma(t) = \int k_{diss}(E)g(E,t)dE / \int g(E,t)dE$.

## Data availability

The data generated in this study have been deposited in the Zenodo database under accession code https://doi.org/10.5281/zenodo.7320010. Source data are provided with this paper.

## Code availability

Data were collected using LabView 2022 and Halcon 12. Data were analyzed using Python 2.7.17 including the packages npTDMS 1.3.0, NumPy 1.16.6, PyAbel 0.8.5, SciPy 1.2.3, and uncertainties 3.1.5. Data were visualized using Python 2.7.17 including the package matplotlib 2.2.5. The code used in this study is openly available in Zenodo at https://doi.org/10.5281/zenodo.7320010.

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

## Acknowledgements

This work was supported by Swedish Research Council grant numbers 2016-03675 (MHS), 2018-04092 (HTS), 2019-04379 (HC), 2020-

03437 (HZ), Knut and Alice Wallenberg Foundation grant number 2018.0028 (HC, HTS, and HZ), Olle Engkvist Foundation grant number 200-575 (MHS), and Swedish Foundation for International Collaboration in Research and Higher Education (STINT) grant number PT2017-7328 (JNB and MHS). We acknowledge the DESIREE infrastructure for provisioning of facilities and experimental support, and thank the operators and technical staff for their invaluable assistance. The DESIREE infrastructure receives funding from the Swedish Research Council under the grant numbers 2017-00621 and 2021-00155. This article is based upon work from COST Action CA18212 - Molecular Dynamics in the GAS phase (MD-GAS), supported by COST (European Cooperation in Science and Technology).

## Author contributions

M.H.S.: Conceptualization, Data curation, Formal Analysis, Funding acquisition, Investigation, Methodology, Project administration, Software, Supervision, Visualization, Writing – original draft. J.N.B.: Conceptualization, Data curation, Formal Analysis, Funding acquisition, Investigation, Project administration, Writing – review and editing. H.C.: Funding acquisition, Resources, Writing – review and editing. S.I.: Investigation, Writing – review and editing. M.C.J.: Investigation, Writing – review and editing. J.E.N.N.: Investigation, Writing – review and editing. H.T.S.: Funding acquisition, Resources, Supervision, Writing – review and editing. H.Z.: Funding acquisition, Resources, Supervision, Writing – review and editing. B.Z.: Investigation, Writing – review and editing.

## Funding

## Competing interests

The authors declare no competing interests.
