## [Peer Review File · Nature Communications]

REVIEWER COMMENTS

Reviewer #1 (Remarks to the Author):

The manuscript by Stockett and co-workers is an experimental and theoretical investigation of the dissociation of cyanonaphthalene cation as a function of internal energy. This is important because cyanonaphthalene was recently unambiguously identified in Space. Despite PAHs being invoked as comprising as much as 20% of cosmic carbon and being responsible for 15% of galactic PAH emission, until this identification no specific PAH has been identified. It was observed at abundances much higher than expected.

In this paper, Stockett and coworkers show that by taking account of recurrent fluorescence, the stabilization of this CNN⁺ cation can be accounted for. Furthermore, it is shown that taking account of Herzberg-Teller coupling is of crucial importance in predicting the correct emission intensity.

Overall, this shows that small PAHs can be stabilised, and lends support to interstellar PAHs being smaller than what has been believed for some time. Because this paper makes these important science points, it is suitable for publication in a high impact journal. That said, it is on the technical side.

I have just a few points.

1. It is said that the transitions to the lowest excited states are symmetry forbidden. This is not true. CNN⁺ is in the Cs point group which has no symmetry forbidden transitions. The parent naphthalene does have sufficient symmetry within the D_{2h} point group such that the lowest transition is gerade-gerade, and thus we can expect the lowest energy transition of CNN⁺ to be weak. But the main reason that these species are not considered DIB candidates (discussion) is that the lowest transitions are too low in energy and the stronger transitions in the visible are broadened by vibronic coupling back to the D₁ state.

2. The recurrent fluorescence mechanism seems identical to the Poincaré Fluorescence coined by Léger, Boissel and d'Hendecourt. Perhaps this should be cited (apologies if I missed it).

<https://journals.aps.org/prl/pdf/10.1103/PhysRevLett.60.921>

3. Trivial matter: the sentence that begins "McGuire et al. indicates..." I take it the singular implied by the verb form indicates that this is the paper rather than the author, but the sentence suggests that the paper limits the abundance of naphthalene. It could be reworded.

Reviewer #2 (Remarks to the Author):

The paper "Resilience of small PAHs in interstellar clouds: Efficient stabilization of cyanonaphthalene by fast radiative cooling" by Stockett and coworkers examines the relaxation pathways of internally excited cyanonaphthalene cations. Unimolecular dissociation and radiative cooling rate coefficients are determined using different measurements in the cryogenic ion storage ring DESIREE.

The key result is that the cyanonaphthalene cation exhibits a much higher radiative relaxation rate, and thus radiative stabilization, than previously thought, owing to Herzberg-Teller coupling that increases the electronic cooling rates through recurrent

fluorescence. The authors relate this behaviour, also seen in other small polycyclic aromatic that were previously thought to be destroyed easily by dissociation upon internal excitation, to the recently astronomically observed high abundance of cyanonaphtalene in the cold dark cloud TMC-1, the first specific PAH detected in the ISM.

This highly interesting result is of broad interest to astrochemists, astronomers, as well as gas-phase chemistry and molecular physics scientists. It questions the common belief that only large PAHs can survive in the interstellar medium and provides a nice example on how laboratory experiments can guide in the interpretation of astronomical observations.

These are high-quality experiments that can only be performed at few experimental platforms around the world under these conditions. The presented data and analysis are conclusive and described in sufficient and well-explained detail, and the authors back up their results with high-level quantum-chemical calculations.

I only have a few minor comments that the authors should consider, and I am confident that after addressing these points this paper will be publishable in Nature Comm.

1. My main scientific issue is with the kinetic model used. As the authors state in the last paragraph of section 4.1, other dissociation channels than the HCN loss channel exist. And these dissociation channels might very likely show a different energy dependence of the rate coefficient owing to different appearance energies. Should this not be accounted for in the Master Equation and KER analysis? I.e., if there is a much faster H-loss channel at higher energies that is not detected in the experiment (only HCN-loss is monitored), should that not change the cooling rate of the excited cyanonaphtalene cations dramatically?

2. In the Discussion, the authors conclude that recurrent fluorescence plays a more important role in the radiative stabilization of small PAHs as previously thought, in particular in the presence of Herzberg-Teller coupling. In my view this is the key messages of the present work, since it changes our view on the chance of survival of small PAHs in the interstellar medium and challenges the GrandPAH hypothesis. In this respect I would like to see a more general discussion on this. The authors make reference to a few other studies on small PAH cations, that support their claim. Thus, my question, which I think might be asked by many astrochemists, is how general this statement really is. Is strong recurrent fluorescence only present in cationic PAHs, or only in neutrals? Does functionalization, e.g., the CN group in the case of cyanonaphtalene, have an influence? I understand that answering this question might involve more laboratory measurements and/or elaborate calculations, but maybe the authors can comment on this in view of their experience on other systems?

3. The authors performed high-level quantum-chemical calculations on the vibrational and electronic transitions of the 1-CNN⁺ ion. I suggest to add the output of these calculations to the supplementary information or the zenodo archive (which I could not access, the link did not work yet), since they are of use to the spectroscopic and observational community.

4. Minor comment: The paper in general is very well written and easy to read. There are, however, quite a few typos (e.g., "aromiatic molecules", "moleular cloud", "an keystone" on the first page) that the autos should correct. Also on the first page, the sentence "While laboratory studies have shown that nitrogen bearing aromatic molecules ... " continues without finishing this half sentence. I assume the authors mean "... aromatic molecules, as well as pure hydrocarbons like naphthalene and indene, can be formed efficiently by gas-phase routes under cold conditions ..." or something like this.

Reviewer #3 (Remarks to the Author):

This article presents new, original measurements of radiative cooling rate coefficients for the 1-cyanonaphtalene cation (1-CNN⁺) using the DESIREE ion storage ring. They find that

1-CNN+ can be efficiently stabilized by recurrent fluorescence, which contradicts the long-standing assumption that small PAHs do not survive in the ISM as they are not able to be stabilized following the absorption of a photon or a collisional event.

This work is highly significant to the field and will help to improve our understanding of the stability of small PAHs in the ISM. This is particularly relevant in the JWST era as well as due to new capabilities to detect small PAHs in the cold ISM at radio wavelengths. The results may also be of wider interest in other areas of chemical physics and combustion chemistry.

I am not sufficiently qualified on the details of the experimental technique to comment on whether or not the methodology is sound or whether the data is interpreted correctly. It does appear that the authors have carefully considered many factors in their analysis, such as the detector efficiency, and have incorporated these into their calculations. I believe the conclusions are well supported by the data. I believe there is sufficient detail for the work to be reproduced.

The authors do not claim that this mechanism would "fix" the discrepancies between the observed and modelled abundances presented by McGuire et al. but suggest that it may be one method of bringing these values closer together. They did not exaggerate their results but presented a nice discussion of their importance, particularly in the case of 1-CNN in TMC-1. I found that they described well how their results could be used to update astrochemical models and these results will be of interest to astrochemical modellers.

In general, I found it a little unclear where the authors were describing processes relevant to TMC-1 or to diffuse regions of space (that could then potentially produce stabilized PAHs that could be inherited in dense clouds). I suggest that the authors try to clarify this as much as possible throughout the manuscript to make it easily accessible to astronomers.

Minor comments:

1. Page 1, paragraph 2. The authors state "as well as that of the smaller but structurally similar benzonitrile, are orders of magnitude higher than predicted by current astrochemical modeling"

However, the reference they link shows only a difference of a factor of 4 here. The reference should be changed or benzonitrile removed as an example.

2. Introduction: "While laboratory studies have shown that nitrogen-bearing aromatic molecules⁴, as well as pure hydrocarbons like naphthalene⁵ and indene^{6,7}, the model of McGuire et al. indicates rapid depletion of CNN from TMC-1 ..."

The first part of this sentence appears incomplete. While laboratory studies have shown what?

3. The role of recurrent fluorescence is inferred indirectly by following the rate of unimolecular dissociation through the product, HCN. Quantifying the RF rate coefficient relies on a number of assumptions that, for someone outside of the area, are difficult to follow. Could the authors add a short paragraph describing the major assumptions and sources of uncertainty? For example, in equation two, are there any other processes that could contribute to $R(t)$?

4. Page 6: 'In summary, the present laboratory study suggests that the situation in TMC-1 is closer to the "best case scenario" discussed by McGuire et al., with high inherited naphthalene abundance and no destruction by ions, than to their baseline assumptions of low inherited naphthalene and ion interactions leading always

I think this should be "best case scenarios" where scenario 1) is high inherited abundance of naphthalene and scenario 2) no destruction by ions. McGuire et al. do not present a model where both naphthalene is inherited and ion destruction turned off. However, it is likely that a combination of the two would better reproduce the observed abundances. The

authors should rephrase this sentence and perhaps add a note about the very high naphthalene abundance that would be needed (as discussed in McGuire et al.)

REVIEWER COMMENTS

Reviewer # 1 (Remarks to the Author):

The manuscript by Stockett and co-workers is an experimental and theoretical investigation of the dissociation of cyanonaphthalene cation as a function of internal energy. This is important because cyanonaphthalene was recently unambiguously identified in Space. Despite PAHs being invoked as comprising as much as 20% of cosmic carbon and being responsible for 15% of galactic PAH emission, until this identification no specific PAH has been identified. It was observed at abundances much higher than expected.

In this paper, Stockett and coworkers show that by taking account of recurrent fluorescence, the stabilization of this CNN+ cation can be accounted for. Furthermore, it is shown that taking account of Herzberg-Teller coupling is of crucial importance in predicting the correct emission intensity.

Overall, this shows that small PAHs can be stabilised, and lends support to interstellar PAHs being smaller than what has been believed for some time. Because this paper makes these important science points, it is suitable for publication in a high impact journal. That said, it is on the technical side.

I have just a few points.

1. It is said that the transitions to the lowest excited states are symmetry forbidden. This is not true. CNN+ is in the Cs point group which has no symmetry forbidden transitions. The parent naphthalene does have sufficient symmetry within the D2h point group such that the lowest transition is gerade-gerade, and thus we can expect the lowest energy transition of CNN+ to be weak. But the main reason that these species are not considered DIB candidates (discussion) is that the lowest transitions are too low in energy and the stronger transitions in the visible are broadened by vibronic coupling back to the D1 state.

Yes, the lowest excited state of CNN+, in contrast to unsubstituted naphthalene, is not strictly forbidden. We have rephrased:

This transition, ~~unlike the first electronic transitions of most highly symmetric PAHs³⁰, is not strictly forbidden by symmetry, but is still quite weak in the Franck-Condon limit~~ ~~vibronic couplings relax this restriction~~. Including H-T coupling, the calculated transition oscillator strength $f = 0.011$ and hence RF rate coefficient (Eq. 14) is two orders of magnitude higher than if vibronic coupling is neglected (see Sec. 4.2). A simulation with

the lower oscillator strength ($f = 10^{-4}$) is shown by the dashed line...

Herzberg-Teller coupling increases the absorption probability of the lowest energy transitions in PAHs, which as you note are not found in the DIBs. We think this further challenges, rather than supports, the PAH-DIB hypothesis. We prefer our original wording which makes no strong claims on the PAH-DIB hypothesis, which is beyond the scope of this investigation, but rather recommends only general caution in comparing calculated absorption spectra to astronomical observations.

2. The recurrent fluorescence mechanism seems identical to the Poincaré Fluorescence coined by Léger, Boissel and d’Hendecourt. Perhaps this should be cited (apologies if I missed it).

<https://journals.aps.org/prl/pdf/10.1103/PhysRevLett.60.921>

Yes, these are two names for the same process. We had cited a different paper by Boissel *et al.* [ref 27 in original manuscript]. We now cite both papers at first mention of Recurrent Fluorescence:

However, this reasoning is based on the assumption that cooling only occurs *via* emission of IR photons from transitions between vibrational levels, **which is indeed slow^{9,10}. However, faster cooling may be possible** due to emission of optical photons from thermally populated electronically excited states, *i.e.* Recurrent Fluorescence ^{11,12} (RF), **also called Poincaré fluorescence.**

3. Trivial matter: the sentence that begins "McGuire et al. indicates..." I take it the singular implied by the verb form indicates that this is the paper rather than the author, but the sentence suggests that the paper limits the abundance of naphthalene. It could be reworded.

Yes, this passage was also noted as problematic by the other reviewers, the new text reads:

Laboratory studies have shown that nitrogen bearing aromatic molecules⁵, as well as pure hydrocarbons like naphthalene⁶ and indene^{4,7}, **may be readily formed in barrierless, gas-phase ion-molecule and radical-molecule reactions under conditions similar to those in interstellar clouds. On the other hand,** the model of McGuire *et al.* indicates rapid depletion of CNN from TMC-1

through interactions with ions,³ and it predicts a limited abundance of its main precursor, naphthalene, which has also been shown to be a keystone in the formation of larger PAHs⁸.

Reviewer # 2 (Remarks to the Author):

The paper “Resilience of small PAHs in interstellar clouds: Efficient stabilization of cyanonaphthalene by fast radiative cooling” by Stockett and coworkers examines the relaxation pathways of internally excited cyanonaphthalene cations. Unimolecular dissociation and radiative cooling rate coefficients are determined using different measurements in the cryogenic ion storage ring DESIREE.

The key result is that the cyanonaphthalene cation exhibits a much higher radiative relaxation rate, and thus radiative stabilization, than previously thought, owing to Herzberg-Teller coupling that increases the electronic cooling rates through recurrent fluorescence. The authors relate this behaviour, also seen in other small polycyclic aromatic that were previously thought to be destroyed easily by dissociation upon internal excitation, to the recently astronomically observed high abundance of cyanonaphthalene in the cold dark cloud TMC-1, the first specific PAH detected in the ISM.

This highly interesting result is of broad interest to astrochemists, astronomers, as well as gas-phase chemistry and molecular physics scientists. It questions the common belief that only large PAHs can survive in the interstellar medium and provides a nice example on how laboratory experiments can guide in the interpretation of astronomical observations.

These are high-quality experiments that can only be performed at few experimental platforms around the world under these conditions. The presented data and analysis are conclusive and described in sufficient and well-explained detail, and the authors back up their results with high-level quantum-chemical calculations.

I only have a few minor comments that the authors should consider, and I am confident that after addressing these points this paper will be publishable in Nature Comm.

1. My main scientific issue is with the kinetic model used. As the authors state in the last paragraph of section 4.1, other dissociation channels than the HCN loss channel exist. And these dissociation channels might very likely show a different energy dependence of the rate coefficient owing to different appearance energies. Should this not be accounted for in the Master

Equation and KER analysis? I.e., if there is a much faster H-loss channel at higher energies that is not detected in the experiment (only HCN-loss is monitored), should that not change the cooling rate of the excited cyanonaphthalene cations dramatically?

Yes, it would be more complete to fully model other dissociation channels. Unfortunately, we are unaware of any published dissociation rate coefficients for these channels, and we can not determine them reliably in our experiment. In the manuscript, the effect of the main competing channel, H-loss, is treated approximately based on the experimental data that is available, namely the branching fraction extracted from published breakdown curves [West 2019]. According to these data, the H-loss channel has a similar appearance energy to, with a lower yield than, HCN-loss. H-loss can be expected to contribute to the depletion of the stored ion beam, which is accounted for in our determination of the detector efficiency. However, H fragments are detected with much lower efficiency ($\sim 2\%$) than HCN and will not significantly contribute to the measured $R(t)$. Due to their low mass, H fragments will also be distributed over a much wider area of the detector than HCN fragments with similar KER. Thus those H fragments that are detected will mainly contribute to the background term in the KER analysis and hence are also accounted for there to some degree. The C_2H_2 -loss contribution is harder to account for, but is expected to be small due to the higher appearance energy (4 eV vs 3 eV for H- and HCN-loss). At the referee's suggestion, we have included the depletion of the beam by H-loss in our kinetic model in the same approximate way it is included in the detector efficiency, by its branching fraction. Our master equation now reads:

$$\begin{aligned} \frac{d}{dt}g(E, t) = & -k_{diss}(E)g(E, t)/Y_{\text{HCN}} \\ & + \sum_s [k_s(E + h\nu_s)g(E + h\nu_s, t) - k_s(E)g(E, t)] \\ & + k_{RF}(E + h\nu_{el})g(E + h\nu_{el}, t) - k_{RF}(E)g(E, t). \quad (1) \end{aligned}$$

The first term gives the depletion of the population by unimolecular dissociation. The factor $Y_{\text{HCN}} = 0.70$ is the branching fraction for HCN-loss estimated from published breakdown curves²² and accounts for depletion of the ion beam by H-loss.

The result of this modification is a small change in the temperature of the best fitting simulation, from 1860(130) K to 1970(60) K. We thank the referee for suggesting this improvement to our model. The possible contribution of other dissociation channels is now also mentioned in the beginning of the results section, rather than at the end of the paper:

During the measurement, the yield of neutral fragments (~~HCN molecules according to Eq~~) leaving the storage ring is recorded continuously as a function of time t after the ions left the source at $t = 0$. The measured count rate, $R(t)$, is averaged over a large number of injection and storage cycles. ~~We assume that $R(t)$ is dominated by dissociation of 1-CNN⁺ yielding HCN molecules according to Eq. 1. The products of competing dissociation channels are not expected to contribute significantly to $R(t)$, and are discussed in Supplementary Note 1.~~

For completeness, we have added a section to the supplementary information (Supplementary Note 1) which presents an alternative kinetic model including estimates of the rate coefficients for H- and C₂H₂-loss. The results of this alternative model are shown to agree with our approximate model well within their uncertainties. We prefer to keep our approximate model in the main text, as we don't want to give the false impression that our results are heavily reliant on data we have not measured ourselves.

There may be additional fragmentation channels opening at higher energy, possibly dominating over HCN-loss. This does not effect the radiative cooling rate coefficients our our determination thereof, as our method is based on the quenching of the lowest-energy channel, here HCN-loss. A faster dissociation channel at higher energies would increase the effective cooling rate of the ensemble at early times ($t \ll k_c^{-1}$), as the high-energy wing of the ensemble would be depleted more rapidly. The presence of competing dissociation channels has been invoked as an explanation for measured decay rates which are best fit with power-law exponents differing from unity *i.e.* $R(t) \propto t^{-p}e^{-k_c t}$ with $p \neq 1$. Such behavior is not observed in the present experiment, and so while we can not rule out contributions from high-energy channels, we have no motivation for speculating about them.

2. In the Discussion, the authors conclude that recurrent fluorescence plays

a more important role in the radiative stabilization of small PAHs as previously thought, in particular in the presence of Herzberg-Teller coupling. In my view this is the key messages of the present work, since it changes our view on the chance of survival of small PAHs in the interstellar medium and challenges the GrandPAH hypothesis. In this respect I would like to see a more general discussion on this. The authors make reference to a few other studies on small PAH cations, that support their claim. Thus, my question, which I think might be asked by many astrochemists, is how general this statement really is. Is strong recurrent fluorescence only present in cationic PAHs, or only in neutrals? Does functionalization, e.g., the CN group in the case of cyanonaphthalene, have an influence? I understand that answering this question might involve more laboratory measurements and/or elaborate calculations, but maybe the authors can comment on this in view of their experience on other systems?

Herzberg-Teller coupling and Recurrent Fluorescence are completely general and present in all poly-atomic molecules regardless of charge state. Whether or not they are decisive in the stability of specific molecules is more subtle. In naphthalene and larger acenes, for example, the lowest energy $D_1 \leftarrow D_0$ transition is forbidden by symmetry. H-T coupling in itself does not remove this constraint, *but substitution does*, as is the case with CNN. The lowest energy transition is still weak in 1-CNN⁺ but is strengthened by H-T coupling, leading to efficient radiative stabilization. On the other hand, RF has been observed to stabilize cationic naphthalene and higher acenes *via* the second lowest transition ($D_2 \rightarrow D_0$). This transition is symmetry allowed and also H-T-enhanced. Even though this transition is higher in energy and thus much less rapid, the dissociation energies of unsubstituted acenes are significantly higher (4-5 eV) compared to cyano- and other substituted PAHs (3 eV for 1-CNN⁺), so RF still plays a role. On yet another hand, we have examples of substituted PAHs which do not show RF [Zhu et al J. Chem. Phys. 157, 044303 (2022)]. We have reworded the relevant paragraph to highlight this important point:

While it has long been known that Herzberg-Teller vibronic coupling influences the optical spectra of PAHs³², the implications are not widely incorporated in astronomical contexts. **For example**, comparisons of measured and simulated optical spectra of PAHs to astronomical observations such as the Diffuse Interstellar Bands often discount the importance of the lowest energy

transitions, on the basis of their low oscillator strengths in the Franck-Condon limit. In the present case, the large enhancement of the oscillator strength of the ~~weaksymmetry forbidden~~ $L_\alpha \leftarrow D_0$ transition leads to efficient radiative stabilization of 1-CNN⁺ by ~~RFrecurrent fluorescence. This effect~~ Generally, H-T coupling and RF are both important for PAHs^{13,14,33,34}, and ~~the possibility of fast radiative cooling~~ challenges the long-held assumption that small PAHs are rapidly depleted from the interstellar medium (ISM) by UV photodestruction^{9,10}. ~~The importance of RF in stabilizing a particular PAH depends sensitively on the absolute dissociation rate coefficients and optical transition probabilities.~~ Calculations of these properties, including H-T coupling, benchmarked to quantitative laboratory data such as those presented here, will be crucial ~~for such comparisons to predicting the stability of PAHs and other complex molecules in space.~~

3. The authors performed high-level quantum-chemical calculations on the vibrational and electronic transitions of the 1-CNN⁺ ion. I suggest to add the output of these calculations to the supplementary information or the zenodo archive (which I could not access, the link did not work yet), since they are of use to the spectroscopic and observational community.

The Zenodo is now available at <http://doi.org/10.5281/zenodo.7320010> and included the output files from our calculations.

4. Minor comment: The paper in general is very well written and easy to read. There are, however, quite a few typos (e.g., “aromatic molecules”, “molecular cloud”, “an keystone” on the first page) that the authors should correct. Also on the first page, the sentence “While laboratory studies have shown that nitrogen bearing aromatic molecules . . . “ continues without finishing this half sentence. I assume the authors mean “. . . aromatic molecules, as well as pure hydrocarbons like naphthalene and indene, can be formed efficiently by gas-phase routes under cold conditions . . .” or something like this.

Thanks, we have fixed these typos and also finished the orphaned clause:

Laboratory studies have shown that nitrogen bearing aromatic molecules⁵, as well as pure hydrocarbons like naphthalene⁶ and indene^{4,7}, ~~may be readily formed in barrierless, gas-phase ion-~~

molecule and radical-molecule reactions under conditions similar to those in interstellar clouds. On the other hand, the model of McGuire *et al.* indicates rapid depletion of CNN from TMC-1 through interactions with ions,³ and it predicts a limited abundance of its main precursor, naphthalene, which has also been shown to be a keystone in the formation of larger PAHs⁸.

Reviewer # 3 (Remarks to the Author):

This article presents new, original measurements of radiative cooling rate coefficients for the 1-cyanonaphthalene cation (1-CNN+) using the DESIREE ion storage ring. They find that 1-CNN+ can be efficiently stabilized by recurrent fluorescence, which contradicts the long-standing assumption that small PAHs do not survive in the ISM as they are not able to be stabilized following the absorption of a photon or a collisional event.

This work is highly significant to the field and will help to improve our understanding of the stability of small PAHs in the ISM. This is particularly relevant in the JWST era as well as due to new capabilities to detect small PAHs in the cold ISM at radio wavelengths. The results may also be of wider interest in other areas of chemical physics and combustion chemistry.

I am not sufficiently qualified on the details of the experimental technique to comment on whether or not the methodology is sound or whether the data is interpreted correctly. It does appear that the authors have carefully considered many factors in their analysis, such as the detector efficiency, and have incorporated these into their calculations. I believe the conclusions are well supported by the data. I believe their is sufficient detail for the work to be reproduced.

The authors do not claim that this mechanism would “fix” the discrepancies between the observed and modelled abundances presented by McGuire *et al.* but suggest that it may be one method of bringing these values closer together. They did not exaggerate their results but presented a nice discussion of their importance, particularly in the case of 1-CNN in TMC-1. I found that they described well how their results could be used to update astrochemical models and these results will be of interest to astrochemical modellers.

In general, I found it a little unclear where the authors were describing processes relevant to TMC-1 or to diffuse regions of space (that could then potentially produce stabilized PAHs that could be inherited in dense clouds). I suggest that the authors try to clarify this as much as possible throughout

the manuscript to make it easily accessible to astronomers.

OK, we have tried to clarify this distinction throughout the paper.

Minor comments: 1. Page 1, paragraph 2. The authors state “as well of that of the smaller but structurally similar benzonitrile, are orders of magnitude higher than predicted by current astrochemical modeling” However, the reference they link shows only a difference of a factor of 4 here. The reference should be changed or benzonitrile removed as an example.

Yes, this passage seems to have been garbled. The corrected text reads:

As important as this assignment is for finally confirming the presence of PAH molecules in space, it is equally remarkable that the observed CNN abundances, ~~as well of that of the smaller but structurally similar benzonitrile,~~ are six orders of magnitude higher than predicted by astrochemical modeling². ~~Notably, the same model generally reproduces the observed abundances of linear and mono-cyclic nitriles³, but underpredicts the abundance of bicyclic indene by four orders of magnitude⁴.~~

2. Introcuton: ”While laboratory studies have shown that nitrogen-bearing aromatic molecules⁴, as well as pure hydrocarbons like naphthalene⁵ and indene^{6,7}, the model of McGuire et al. indicates rapid depletion of CNN from TMC-1 ...”

The first part of this sentence appears incomplete. While laboratory studies have shown what?

Again we apologize for the mangled paragraph, which now reads:

Laboratory studies have shown that nitrogen bearing aromatic molecules⁵, as well as pure hydrocarbons like naphthalene⁶ and indene^{4,7}, ~~may be readily formed in barrierless, gas-phase ion-molecule and radical-molecule reactions under conditions similar to those in interstellar clouds. On the other hand,~~ the model of McGuire *et al.* indicates rapid depletion of CNN from TMC-1 through interactions with ions,³ ~~and it predicts a limited~~ abundance of its main precursor, naphthalene, which has also been shown to be a keystone in the formation of larger PAHs⁸.

3. The role of recurrent fluorescence is inferred indirectly by following the rate

of unimolecular dissociation through the product, HCN. Quantifying the RF rate coefficient relies on a number of assumptions that, for someone outside of the area, are difficult to follow. Could the authors add a short paragraph describing the major assumptions and sources of uncertainty? For example, in equation two, are there any other processes that could contribute to $R(t)$?

Indeed, we quantify the RF rate indirectly, and thus the main source of uncertainty is the rate coefficient of the channel we monitor directly: dissociation. An important advance in the present work is that we determine the dissociation rate coefficient within the same experiment, rather than using literature values which often have large uncertainties. We highlight this now in the introduction:

...most reports of RF have been inferred indirectly through the quenching of dissociation or other destruction channels on timescales that are too rapid to be explained by sequential emission of IR photons^{13,17,18}. Indirect determinations of RF rates depend sensitively on the absolute rate coefficient of the monitored destruction channel, which often have been calculated using simple statistical models^{18,19} or determined through separate mass spectrometry experiments with much shorter sampling times^{12-14,20}. Additionally, it is not uncommon for published rate coefficients for the same molecule and destruction channel to differ by orders of magnitude^{12,21}. Here, we present a method to determine unimolecular dissociation and RF quenching rate coefficients within the same experiment.

The key assumptions in determining the dissociation rate coefficient are found in Equations 8 and 9 in the main text. In Eq. 8, we assume $A^{diss} \approx \frac{k_B T^\ddagger}{h}$, eliminating the parameter ΔS^\ddagger from the model. This is not really necessary as we present in a new section in the SI (Supplementary Note 2) where we show that $1 + \Delta S^\ddagger / N_A k_B \approx 0$. Eliminating ΔS^\ddagger simply improves our estimates of ΔT^\ddagger by removing the covariance between these parameters. It does not significantly effect the final determination of E_a or its uncertainty. Note that the original text erroneously stated this assumption as $\Delta S^\ddagger \approx 0$. The main text now points to the SI and gives the correct expression:

Given such a ‘flat’ transition state²⁶, we invoke the simplifying assumption $1 + \frac{\Delta S^\ddagger}{N_A k_B} \approx 0$ (see Supplementary Note 2), giving...

In Eq. 9, we claim that the parameter r_0 , which holds all the instrumental factors and the fraction of ions having the temperature ΔT^\ddagger , is the same number we find from the fit to the total decay rate $R(t)$. This is true assuming that the HCN-loss process we monitor is the only one contributing to $R(t)$. This question is addressed in a new section in the supplementary information (Supplementary Note 2); see also the response to Reviewer 2. We have attempted to clarify this point in our discussion of Eq. 9:

... where Δt is the camera exposure time, and $k_{diss}(E(T^\ddagger))$ is given by Eq. 8. The value of r_0 is taken from the fit to $R(t)$, which is to assume that $R(t)$ is dominated by the modeled HCN-loss channel (see Supplementary Note 1). The values of $\Gamma(t)$ extracted from the imaging data (Eq. 9) are shown in Fig. 2, labeled ‘ $\Gamma(t)$ Img.’, and agree well with those from the decay rate measurement (Eq. 4), supporting this assumption. The fit of $P(\epsilon, t)$ includes contributions...

Beyond these key assumptions, which are now highlighted more clearly in the revised manuscript and justified more fully in the new Supplementary Notes, the determination of k_{diss} and thus the RF rate is limited only by measurement quality.

4. Page 6: ‘In summary, the present laboratory study suggests that the situation in TMC-1 is closer to the “best case scenario” discussed by McGuire et al., with high inherited naphthalene abundance and no destruction by ions, than to their baseline assumptions of low inherited naphthalene and ion interactions leading always

I think this should be “best case scenarios” where scenario 1) is high inherited abundance of naphthalene and scenario 2) no destruction by ions. McGuire et al. do not present a model where both naphthalene is inherited and ion destruction turned off. However, it is likely that a combination of the two would better reproduce the observed abundances. The authors should rephrase this sentence and perhaps add a note about the very high naphthalene abundance that would be needed (as discussed in McGuire et al.)

Yes, what we call the best case scenario was not explicitly considered in McGuire *et al.* We have better explained ourselves in the revised text:

In summary, laboratory studies, including the present one, suggest that the underprediction of the CNN abundance in TMC-1 may be partially rationalized by a combination of both of the alternative scenarios considered by McGuire *et al.*: high inherited naphthalene abundance and no destruction by ions.² In their model, each of these assumptions individually reduced the discrepancy by about two orders of magnitude², although it should be noted that the assumed inherited naphthalene abundance was on the order of 1% of the available carbon. Additional laboratory studies of ion-neutral, electron-ion, and ion-ion reactions involving CNN under astrophysically relevant conditions should be undertaken to further constrain astrochemical models.

REVIEWERS' COMMENTS

Reviewer #2 (Remarks to the Author):

I am very satisfied how the authors have incorporated my comments, and those of the other reviewers, both in their answer letter and in the manuscript (and SI).
I recommend the manuscript for publication as is.